# Melanoma and Nevus Skin Lesion Classification Using Handcraft and Deep Learning Feature Fusion via Mutual Information Measures

**DOI:** 10.3390/e22040484

**Published:** 2020-04-23

**Authors:** Jose-Agustin Almaraz-Damian, Volodymyr Ponomaryov, Sergiy Sadovnychiy, Heydy Castillejos-Fernandez

**Affiliations:** 1Instituto Politecnico Nacional, Santa Ana Ave. # 1000, Mexico City 04430, Mexico; jalmarazd1401@alumno.ipn.mx; 2Instituto Mexicano del Petroleo, Lazaro Cardenas Ave. # 152, Mexico City 07730, Mexico; 3Academic Area of Computer and Electronics, Institute of Basic Sciences and Engineering, Universidad Autonoma del Estado de Hidalgo, Pachuca–Tulancingo Highway Km. 4.5, Mineral de la Reforma, Hidalgo 42083, Mexico

**Keywords:** fusion, handcraft, deep learning, melanoma, convolutional neural networks, transfer learning, computer-aided systems, mutual information, balance, data

## Abstract

In this paper, a new Computer-Aided Detection (CAD) system for the detection and classification of dangerous skin lesions (melanoma type) is presented, through a fusion of handcraft features related to the medical algorithm ABCD rule (Asymmetry Borders-Colors-Dermatoscopic Structures) and deep learning features employing Mutual Information (MI) measurements. The steps of a CAD system can be summarized as preprocessing, feature extraction, feature fusion, and classification. During the preprocessing step, a lesion image is enhanced, filtered, and segmented, with the aim to obtain the Region of Interest (ROI); in the next step, the feature extraction is performed. Handcraft features such as shape, color, and texture are used as the representation of the ABCD rule, and deep learning features are extracted using a Convolutional Neural Network (CNN) architecture, which is pre-trained on Imagenet (an ILSVRC Imagenet task). MI measurement is used as a fusion rule, gathering the most important information from both types of features. Finally, at the Classification step, several methods are employed such as Linear Regression (LR), Support Vector Machines (SVMs), and Relevant Vector Machines (RVMs). The designed framework was tested using the ISIC 2018 public dataset. The proposed framework appears to demonstrate an improved performance in comparison with other state-of-the-art methods in terms of the accuracy, specificity, and sensibility obtained in the training and test stages. Additionally, we propose and justify a novel procedure that should be used in adjusting the evaluation metrics for imbalanced datasets that are common for different kinds of skin lesions.

## 1. Introduction

Skin cancer has become one of the deadliest diseases for human beings. Globally, each year, between two and three million non-melanoma (less aggressive) cases occur, and over 130,000 melanoma (aggressive) types are diagnosed [1].

Melanoma is the deadliest type of skin cancer. Australia has the highest rates of skin cancer in the world. In 2018, melanoma accounted for about 22% of skin cancer diagnoses, and non-melanoma tumors accounted for about 78% [2]. Studies have shown that this disease is caused most of the time by exposure to UV radiation in daylight, tanning on sunbeds, and skin color, among others. Physicians have suggested that the best way to detect a malignant skin lesion of any kind is early detection. The rate of survival increases to almost 99% over five years if the disease is spotted in the early stages.

Dermoscopy or Epiluminicence Microscopy (ELM) is a medical method that helps a physician to recognize if a skin lesion belongs to a benign or malignant type of the disease. This method uses a dermatoscope, a tool that consists of a light source and amplification lens to enhance the view of medical patterns such as ramifications, globs, pigmented networks, veils, and colors, among others.

Since the image processing techniques were developed, Computer-Aided Detection (CAD) systems and approaches in the classification [3,4,5,6,7] and segmentation [8] of a Pigmented Skin Lesion (PSL) have been improved, benefiting patient diagnoses in early stages of the disease without shocking or painful medical procedures.

In this work, we propose a novel approach in the detection of a skin lesion among melanoma or nevus types, using handcraft features that depend on shape, color, and texture, which represent the ABCD rule (Asymmetry Borders-Colors-Dermatoscopic Structures), and combining them with deep learning features; these latter features were extracted using the transfer learning method as a generic feature extractor. At the next step, the most important features according to the Mutual Information (MI) metric features should be selected using the fusion technique, aiming at the best performance by taking into account the influences of both sets of features on the binary classification result.

This paper is organized as follows: Section 2 presents a brief review of the methods used in CAD developments with fused features and their nature, Section 3 explains in detail the proposed method, the materials used, and the evaluation metrics employed, and Section 4 describes the experimental results and presents a brief discussion. The conclusions are detailed in Section 5.

## 2. Literature Survey

Medical detection algorithms are one of the first tools used for determining whether a skin lesion is malignant or benign [9,10,11,12,13]. Nachbar et al. [9] developed a subjective method based on the visual perception of the lesion. ABCD Rule is based on color, shape, and particular structures that appear on skin lesions. Due to the simplicity of the algorithm, it is one of the most practiced for evaluating a lesion by a naked eye exam or using a dermatoscope.

The ABCD medical algorithm is composed of the following parts:Asymmetry *A*: The lesion is bisected into two perpendicular axes at 90° of each other, so as to yield the lowest possible asymmetry score. In other words, whether the lesion is symmetrical or not is determined. For each axis, where asymmetry is found, one point is added.Borders *B*: The lesion is divided into slices by eight axes determining whether a lesion has abrupt borders. If one segment presents an abrupt border, one point is added.Colors *C*: The lesion can contain one or more of the following colors: white, brownish, dark brown, black, blue, and red. They are generated by vessels and melanin concentrations, so for each one color founded, one point per color is added.Dermatoscopic Structures *D*: The lesion has the appearance of the following structures: dots, blobs, pigmented networks, and non-structured areas. A point is added for each structure spotted on the lesion.

The described features are weighted as follows:(1)TDS=1.3∗A+0.1∗B+0.5∗C+0.5∗D,
where TDS is the Total Dermatoscopic Score; if it is less than 4.75, it is concluded that the lesion is benign; if the score is between 4.75 and 5.45, the lesion is considered suspicious; if it is more than 5.45, then it is considered malignant.

This algorithm has subversions, where some elements such as dermatoscopic structures are changed by diameter or darkness [14,15]. Additionally, in [16,17], the addition of features known as EFG properties has been suggested; E stands for elevation or evolution, F for firmness, and G for growth. These features work as complementary information obtained from a PSL. Modifications in the ABCD rule are based on simplifying the evaluation of a skin lesion such that anyone can evaluate themselves and record any change in the injury. If the lesion is not identified by these methods, physicians are obliged to initiate invasive methods such as a biopsy to determine their type.

Adjed et al. [18] proposed a method where the aim is the fusion of structural features using Curvelet and Wavelet transform employing the Fast Digital Curvelet Transform (FDCT) wrapping method, and statistical metrics and texture features such as local binary pattern are then computed. They fused around 200 features via concatenation using the PH2 dataset [19].

Hagerty et al. [20] developed a fusion method where deep features are extracted from images using transfer learning method based on the ResNET-50 Convolutional Neural Network (CNN) architecture. However, the question as to which handcraft features are used in their method is not clear. Moreover, they used a feature selection algorithm, in this case, the $\chi^{2}$ method, for performance revision, employing two datasets: the private set and the second set (a modified version of the ISIC 2018 dataset) [21].

Li et al. [22] used a deep learning approach with the fusion of clinical criteria representations, where as a classifier and fusion method, a boosting tree-learning algorithm called LightGBM is used [23]. This method is applied for color properties (RGB and HSL features), texture properties (SIFT and LBP), and shape properties (solidity and circularity, image ratio, and area ratio). The deep learning features were obtained using the transfer learning method based on the ResNET-50 and DenseNET-201 CNN architectures. Data pertaining to 566 features were processed using the ISIC 2018 dataset [21].

Abbas and Celebi [24] proposed a CAD system where the lesion is processed by a Stack-Based Auto-Encoder (SAE), extracting the deep features from the pixels of a lesion while minimizing the information loss. The handcraft features are extracted for color (the Hill climbing algorithm (HCA)) and for texture (the speed-up robust features (SURF)). Applying a feature fusion approach, they used Principal Component Analysis (PCA), and in the concluding stage, Recurrent Neural Networks (RNN) and A Softmax Linear Classifier were employed.

Among the reviewed methods, most of them use handcraft features and deep learning features with the help of the transfer learning method based on well-known CNN architectures [25,26,27,28,29,30,31]. As one can see, the revised schemes tried to fuse the information extracted from the lesion images, gathering data via the concatenation of the feature vectors, classifiers, and feature selection. The main drawback of such methods is the unawareness of the medical information, which is relevant to physicians besides the extracted data from image processing algorithms. The analyzed methods above employ several possibilities in the fusion of features, but most of them do not consider the importance of each extracted feature according to their nature, which is relevant for a pertinence class. Moreover, some of them lack features based on medical algorithms due to their assumption of the weakness of the perceptual handcraft features based on the subjective visual human system. Modern image processing techniques and machine learning approaches are able to learn the patterns and implement those features, as in a vision scoring system. Finally, some of the reviewed methods attempt to perform a multiclass classification, where the system classifies a lesion image to a specific lesion category. Nevertheless, here a problem endures in the multiclass classification, because the data available for each class are limited, and some of the public databases are not well balanced to perform this classification. As a result, a designed system can perform incorrect classifications. Summarizing, we consider the importance of developing an intelligent system that is able to perform the correct classification of melanoma disease employing both types of features, where medical features have relevance on the classification with the aid of deep learning features, aiming for the best performance.

The novel method considers relevant information obtained from handcraft and deep learning features, improving performance quality presented by commonly used criteria: accuracy, specificity, and sensibility. Different from other schemes, our novel framework encourages the use of ABCD rule features, also known as perceptual features, with a set of features equivalent to or based on a similar medical nature.

### Principal Contributions

The principal contributions of our novel approach in the classification of dermoscopic lesions are summarized as follows:A brief survey of computer-aided detection methods that employ fusion between handcraft and deep learning features is presented.Despite the new tendencies of avoiding the ABCD medical algorithm or any of its variations, we utilized descriptors based on them, such as shape, color, and texture, as a new aggregation, and the extraction of deep learning features were used afterwards.A balanced method was employed due to the inconsistency of the ISIC database with respect to classes. A SMOTE oversampling technique was applied, which in this work demonstrates an improvement in performance at the differentiation of melanoma and benign lesion images.A fusing method that employs relevant mutual information obtained from handcraft and deep learning features was designed, and it appears to demonstrate better performance in comparison with state-of-the-art CAD systems.

## 3. Materials and Methods

In this section, the proposed system is described. A brief conceptual block diagram of the system is illustrated in Figure 1. As an initial step, the pigmented skin lesion image is segmented from surrounding normal skin tissue and artefacts such as veils, hairs, and air bubbles, among others, by color space transformation, mean thresholding, and extraction of the Region of Interest (ROI). Subsequently, using the binary mask image, the ROI image and a set of handcraft features based on shape, color, and texture are extracted. Thereafter, deep learning features are obtained using a selected CNN architecture, which is pre-trained in an Imagenet classification task; this CNN is employed as a feature extractor. All extracted features are concatenated in one vector, which later is fused according to the MI criterion. The selected classifier is trained on the ISIC dataset comprised of both malignant and other benign skin lesion images. Finally, the trained classifier model is used to predict each unseen pigmented skin lesion image as a benign or malignant lesion. The details of each stage of the proposed method are described in the remainder of this section.

### 3.1. Preprocessing

An image I(x,y) that is analyzed can contain a lesion with some artefacts such as veils, hairs, stamps, among others. In the first step, we apply a preprocessing stage, where an image is enhanced [8,32,33].

A Gaussian filter is applied, and this filter is used to blur the artefacts contained on the image as primary targets: hair, marks, and spots, among others. This maintains the geometric/shape form of the lesion.

The Gaussian filter is denoted as follows:(2)$$G\left(x,y\right)=\frac{1}{2\pi \sigma^{2}}\exp\left(-\frac{x^{2}+y^{2}}{2\sigma^{2}}\right),$$
where $\sigma^{2}$ is the variance of the spatial kernel, this step is shown in Figure 2.

CIEL*a*b* space is characterized to more closely approximate the human perception system, where there are channels: L stands for lightness; a* and b* stand for chroma channels, where a* is a parametric measure between magenta and green, and b* is a parametric measure between blue and yellow. The L channel presents values between [0,100], and a* and b* chroma channels have values between [−30,30]. This transformation is used to avoid the correlation between channels, while keeping the perceptual data intact, such as a pigmented skin lesion that is darker than healthy skin tissue, as one of the sub-variants of the ABCD algorithm states [14].

In each channel of CIEL*a*b* for images $I_{L},I_{a},I_{b}$, the mean thresholding procedure is applied. Such thresholding allows one to differentiate skin tissue from lesion tissue. In Figure 3, one can see how the CIEL*a*b* space is able to visually separate this information. These mean values are calculated as follows:(3)$$\bar{I_{L}}=\sum_{x=1}^{m}\sum_{y=1}^{n}\frac{1}{mn}I_{L}\left(x,y\right),$$
(4)$$\bar{I_{a}}=\sum_{x=1}^{m}\sum_{y=1}^{n}\frac{1}{mn}I_{a}\left(x,y\right),$$
(5)$$\bar{I_{b}}=\sum_{x=1}^{m}\sum_{y=1}^{n}\frac{1}{mn}I_{b}\left(x,y\right),$$
where (x,y) are the spatial coordinates, m,n are the sizes of an image, and $\bar{I_{L}},\bar{I_{a}},\bar{I_{b}}$ denote the mean values. The thresholding operation is applied in each channel of the CIEL*a*b* space, forming the thresholded channel images as follows: (6)$$I_{Th_{L}}\left(x,y\right)=\{\begin{array}{ll} 1, & I_{L}\left(x,y\right)\geq \bar{I_{L}},\\ 0, & otherwise, \end{array}$$
(7)$$I_{Th_{a}}\left(x,y\right)=\{\begin{array}{ll} 1, & I_{a}\left(x,y\right)\geq \bar{I_{a}},\\ 0, & otherwise, \end{array}$$
(8)$$I_{Th_{b}}\left(x,y\right)=\{\begin{array}{ll} 1, & I_{b}\left(x,y\right)\geq \bar{I_{b}},\\ 0, & otherwise. \end{array}$$

Afterwards, the following logic operation is applied on each binarized image, $I_{Th_{L}},I_{Th_{a}}$, and $I_{Th_{b}}$, to form a binary mask $I_{bin}\left(x,y\right)$ of the image.
(9)$$I_{bin}\left(x,y\right)=I_{Th_{L}}\cap I_{Th_{a}}\cap I_{Th_{b}}.$$

Example of extracted Binary Masks images are given in Figure 4. Finally, a median filter with a kernel 5×5 is applied to the $I_{bin}\left(x,y\right)$, removing the remaining artefacts, which resists thresholding. Next, a bounding box algorithm is performed. The bounding box [34] is a method that is used to compute an imaginary rectangle that completely encloses the given object. This rectangle can be determined by the x- and y-axis coordinates in the upper-left and lower-right corners of a shape. This method is commonly used in object detection tasks because it estimates the coordinates of the ROI in an image.

Bissotto et al. [35] have shown the effect of the bias between different types of image segmentation, where those biases can negatively affect the performance of classification models. They consider that the usage of the bounding box algorithm to segment a lesion is appropriate because a CNN architecture can extract all the relevant features of a lesion and distinguish it from the surrounding healthy skin. Therefore, we consider this solution to reduce the bias of the classification model before processing.

### 3.2. Handcraft Features

The ABCD rule represents a set of perceptual features stated by the findings of patterns in PSLs. The ABCD method employs features that are based mostly on shapes, color, and texture. The selected features in this study are the representation of medical attributes using image processing algorithms.

Sirakov et al. [36] proposed a method to estimate the asymmetry of a lesion based on the binary mask $I_{bin}$, which is obtained from the previous thresholding step. Then, by rotating it through 180°, the symmetry mask $S_{I_{bin}}$ is formed, and the synthetic image *A* is calculated as follows: (10)$$A=I_{bin}\cup S_{I_{bin}}$$
where *A* is the generated image, which contains the non-overlapping regions of the lesion called false symmetry FS, therefore applying
(11)$$Sym_{0{}^{\circ}}=1-\left(FS/A\right),$$
and this technique is applied on the 0° axis of the binary image.

In this study, a variation of the previous method is proposed, where from the *A* generated image is rotated from the major axis and the minor axis, by applying the same procedure again and finally, computing the average symmetry value between the two axes, as follows: (12)$$Symmetry=\frac{Sym_{0{}^{\circ}}+Sym_{90{}^{\circ}}}{2}.$$
The symmetry values belong to the interval [0, 1], where, if this index approaches the highest value (1), a lesion is more symmetric. The Figure 5 shown the extracted ROI images.

#### 3.2.1. Shape Features

Shape features or geometric features [34] can describe an object or a form in numerical values to represent human perception.

For shape features, the following equations are employed:(13)$$Area=\sum_{x=1}^{m}\sum_{y=1}^{n}I_{bin}\left(x,y\right),$$
where m,n are the sizes of the image, and x,y are the spatial coordinates; therefore, the area consists of the amount of pixels contained in the ROI of a lesion.
(14)$$Perimeter=\sum_{i=1}^{m}\sqrt[{2}]{\left(x_{1}-x_{i-1}\right)+\left(y_{1}-y_{i-1}\right)},$$
where (x,y) are the spatial coordinates of the i−th pixel, which constructs the contour of the region, and the perimeter contains the amount of pixels around the ROI of a lesion.
(15)$$Circularity=\frac{4\pi \cdot Area}{Perimeter^{2}},$$
where the circularity shows the similarity between a shape and a circle.
(16)$$Diameter=\sqrt[{2}]{\frac{1}{2}\left(\mu_{2,0}+\mu_{0,2}\right)\pm \sqrt[{2}]{4\mu_{1,1}^{2}-{\left(\mu_{2,0}-\mu_{0,2}\right)}^{2}}},$$
where the diameter is formed by obtaining the length of the major axis and the minor axis of the shape, computed from the 2nd central moment. This measure connects two pairs of points on the perimeter of the shape.
(17)$$Eccentricity=\frac{{\left(\mu_{0,2}-\mu_{2,0}\right)}^{2}+4\mu_{1,1}}{A},$$
which measures the aspect ratio of the length of the major axis to the length of the minor axis.

#### 3.2.2. Colour Features

Medical algorithms, in particular the ABCD rule, try to present, as features, a set of colors contained on a PSL. Therefore, these features can be replaced by statistical characteristics obtained from color spaces. In this study, the following characteristics are used:(18)$$Min_{ch}=min\left[I_{ch}\left(x,y\right)\right],$$
(19)$$Max_{ch}=max\left[I_{ch}\left(x,y\right)\right],$$
(20)$$Var_{ch}=var\left[I_{ch}\left(x,y\right)\right],$$
(21)$$Mean_{ch}=\bar{\left[I_{ch}\left(x,y\right)\right]},$$
where $I_{ch}\left(x,y\right)$ is the image of a chosen channel for the PSL image in RGB and CIEL*a*b* color spaces.

#### 3.2.3. Texture Features

Haralick et al. [37] proposed the Gray Level Co-occurrence Matrix (GLCM). This method analyzes the statistical texture features of an image. The texture features provide information about how the gray intensities of the PSL of the image are distributed. GLCM shows how often a gray level occurs at a pixel located in a fixed position, using $P_{d}\left(i,j\right)$ as the (i,j) element of the normalized GLCM; $N_{g}$ is the number of gray levels; $\sigma_{x},\sigma_{y}$ and $\mu_{x},\mu_{y}$ are the standard derivations and the mean values among the *i* and *j* axes of the GLCM, and are expressed as follows:(22)$$\mu_{x}=\sum_{i=1}^{N_{g}}\sum_{j=1}^{N_{g}}iP_{d}\left(i,j\right),$$
(23)$$\mu_{y}=\sum_{i=1}^{N_{g}}\sum_{j=1}^{N_{g}}jP_{d}\left(i,j\right),$$
(24)$$\sigma_{x}=\sqrt{\sum_{i=1}^{N_{g}}\sum_{j=1}^{N_{g}}{\left(i-\mu_{x}\right)}^{2}P_{d}\left(i,j\right)},$$
(25)$$\sigma_{y}=\sqrt{\sum_{i=1}^{N_{g}}\sum_{j=1}^{N_{g}}{\left(j-\mu_{y}\right)}^{2}P_{d}\left(i,j\right)}.$$

The 13 features used in this study are as follows:(26)$$ASM=\sum_{i=1}^{N_{g}}\sum_{j=1}^{N_{g}}P_{d}^{2}\left(i,j\right).$$

The angular second moment measures consistency of the gray local values.
(27)$$Contrast=\sum_{i=1}^{N_{g}}\sum_{j=1}^{N_{g}}{\left(i-j\right)}^{2}P_{d}\left(i,j\right),$$

This is the second moment. This characteristic measures the variations between pixels.
(28)$$Correlation=\sum_{i=1}^{N_{g}}\sum_{j=1}^{N_{g}}P_{d}\left(i,j\right)\frac{\left(i-\mu_{x}\right)\left(j-\mu_{y}\right)}{\sigma_{x}\sigma_{y}}.$$

This is the linear dependency of the gray level values.  
(29)$$Variance=\sum_{i=1}^{N_{g}}\sum_{j=1}^{N_{g}}{\left(i-\mu \right)}^{2}P_{d}\left(i,j\right).$$

This is the second moment. It shows the spread around the mean in the surrounding neighborhood.
(30)$$IDM=\sum_{i=1}^{N_{g}}\sum_{j=1}^{N_{g}}\frac{1}{1+{\left(i-j\right)}^{2}}P_{d}\left(i,j\right).$$

This is the Inverse Difference Moment (IDM). It shows how close the elements of the GLCM are in their distribution.
(31)$$Entropy=-\sum_{i=1}^{N_{g}}\sum_{j=1}^{N_{g}}P_{d}\left(i,j\right)\cdot ln\left[P_{d}\left(i,j\right)\right].$$

This is the measure of randomness of the gray values in an image.

Additional texture features that are used in this study are based on the difference statistics using the probability $P_{x-y}\left(k\right)$ that can be written as follows:(32)$$P_{x-y}\left(k\right)=\sum_{i=1}^{N_{g}}\sum_{j=1}^{N_{g}}P_{d}\left(i,j\right),\;k=0,1,\ldots ,N_{g}-1,$$
where $P_{d}\left(i,j\right)$ is the (i,j)th element contained in the GLCM, and $N_{g}$ is the number of gray levels.
(33)$$Sum\;Variance=\sum_{k=2}^{2N_{g}}{\left(k-\mu_{x+y}\right)}^{2}P_{x+y}\left(k\right),$$
(34)$$Sum\;Entropy=-\sum_{k=2}^{2N_{g}}P_{x+y}\left(k\right)log\left[P_{x+y}\left(k\right)\right],$$
(35)$$Difference\;Variance=\sum_{k=0}^{N_{G}-1}{\left(k-\mu_{x-y}\right)}^{2}P_{x-y}\left(k\right),$$
(36)$$Difference\;Entropy=-\sum_{k=0}^{N_{g}-1}P_{x-y}\left(k\right)log\left[P_{x-y}\left(k\right)\right],$$
(37)$$IMCorr_{1}=\frac{H\left(XY\right)-H\left(XY_{1}\right)}{max[H(X)H(Y)]},$$
(38)$$IMCorr_{2}=\sqrt{1-\exp\left\{-2\left[H\left(XY_{2}\right)-H\left(XY\right)\right]\right\}},$$
where H(X),H(Y),H(XY),H(XY1) and H(XY2) are denoted as follows:(39)$$H\left(X\right)=-\sum_{i=1}^{N_{g}}P_{x}\left(i\right)\cdot log\left[P_{x}\left(i\right)\right],$$
(40)$$H\left(Y\right)=-\sum_{i=1}^{N_{g}}P_{y}\left(i\right)\cdot log\left[P_{y}\left(i\right)\right],$$
(41)$$H\left(XY\right)=-\sum_{i=1}^{N_{g}}P_{d}\left(i,j\right)\cdot log\left[P_{d}\left(i,j\right)\right],$$
(42)$$H\left(XY1\right)=-\sum_{i=1}^{N_{g}}\sum_{j=1}^{N_{g}}P_{d}\left(i,j\right)\cdot log\left[P_{x}\left(i\right)\cdot P_{y}\left(j\right)\right],$$
(43)$$H\left(XY2\right)=-\sum_{i=1}^{N_{g}}\sum_{j=1}^{N_{g}}P_{x}\left(i\right)\cdot P_{y}\left(j\right)\cdot log\left[P_{x}\left(i\right)\cdot P_{y}\left(j\right)\right].$$

### 3.3. Deep Learning Features

Based on the discrete convolution operation,
(44)$$W\left(i,j\right)=\left(K*I\right)\left(i,j\right)=\sum_{m}\sum_{n}I\left(i-m,j-n\right)K\left(m,n\right),$$
the CNN [38,39] represents one type of method, namely, deep learning strategies [40], the basis of which is to obtain the information of an image I(i,j), using filters K(m,n), which are trained on a neural network as feed-forward and back-propagation, according to
(45)y=z(W∗X)+b,
where *W* are the computed values for the filters, *z* is the activation function, *X* is the input, and *b* is the bias [40,41].

The design of an architecture of a CNN is a rather complex task due to the statement of parameters such as the number and size of the filters, and the depth, even those that are task-related.

#### 3.3.1. Transfer Learning

The main problem of using a deep learning approach is that a large amount of data is needed to train the network from scratch. Usually, to overcome this problem, the transfer learning method [42,43,44] is applied.

Transfer learning is a technique that can be defined as the generalization of a target task based on applied knowledge extracted from one or more source tasks [43]. The idea originates from human thinking: we do not learn exactly how to recognize a chair, a plane, or a book; we start by recognizing colors, shapes, and textures, and someone else then tells us how to differentiate a chair from an apple. This sharing of knowledge between beings helps us to understand the world as infants. Another idea is handling information collected for a task to solve related ones. Therefore, transfer learning can be defined as follows:

Assume a domain *D*, which consists of two components:(46)D={χ,P(X)},
where χ is a feature space, and there is a marginal distribution: $P\left(X\right),X=\left\{x_{1},\ldots ,x_{n}\right\},x_{i}\in \chi$.

Given a task *T* with two components,
(47)$$T=\left\{\gamma ,P\left(Y|X\right)\right\}=\left\{\gamma ,\eta \right\};Y=\left\{y_{1},\ldots ,y_{n}\right\},y_{i}\in \gamma ,$$
where γ is a label space with a predictive function; η is trained from $\left(x_{i},y_{i}\right),x_{i}\in \chi ,y_{i}\in \gamma$, for each feature vector in the *D* domain; η predicts the corresponding label $\eta \left(x_{i}\right)=y_{i}$.

The paper [43] states self-called scenarios:

Given a source domain and an objective domain $D_{S}$ & $D_{T}$, where D={χ,P(X)} and the related tasks are $T_{S}$ & $T_{T}$, where T={γ,P(Y|X)}, the conditions can vary as follows:$\chi_{S}\neq \chi_{T}$: Feature spaces are different.$P\left(X_{S}\right)\neq P\left(X_{T}\right)$: The marginal probabilities of the distributions are different.$\gamma_{S}\neq \gamma_{T}$: The label spaces are different.$P\left(Y_{S}|X_{S}\right)\neq P\left(Y_{T}|X_{T}\right)$: The conditional probabilities are different.

Transfer learning is defined in [44]. Given a source domain $D_{S}$ with a corresponding source task $T_{s}$ and a target domain $D_{T}$ with a corresponding task $T_{T}$, transfer learning is the process of enhancement for a target predictive function $f_{T}\left(\cdot \right)$ using the related information from the domain source $D_{S}$ and the task-related $T_{S}$.

CNN architectures are overlay filters that sample the data contained in an image. Therefore, these filters are hierarchical representations called feature maps when all filters learn all the features based on image data. These are connected to the last layer of the CNN architecture, which is a neural network classifier referred to in the literature as a fully connected layer [40].

Moreover, CNN architectures belong to a class of inductive learning algorithms, where the objective of these algorithms is to map input features between classes seeking the generalization of the data.

Therefore, inductive learning can be transferred from an architecture trained on the source task to the target class, and this is done by adjustments of the model space, correcting the inductive bias. Commonly, this can be performed by replacing the last layer of the model, which is the classifier, from the original one to a lightweight classifier, which should be trained on the generalized features.

In this study, we employed the transfer learning method on architectures that are pre-trained on a similar task. In this research, the Imagenet classification task [45] was employed, by the assumption that $D_{S}=D_{T}$ and the CNN Architecture should perform as a generic feature extractor.

#### 3.3.2. Feature Extractor

In our case, the CNN Architecture was used as a deep feature extractor, where features are extracted by the following:

Consider an image I(x,y) of the domain $D_{T}$ that is mapped or transformed by
(48)$$W=\left\{w_{1},\ldots ,w_{n}\right\};w_{1}\;\in R^{M\times N\times h},$$
where *W* is the weight computed by the feature extractor, where M,N, and *h* are the proposed size of the CNN architecture. As a result, we can obtain
(49)$$P=W\left(I\left(x,y\right)\right)=\left\{w_{1}\left(I\left(x,y\right)\right),\ldots ,w_{n}\left(I\left(x,y\right)\right)\right\};\;\in R^{M\times N\times h},$$
and the pooling transformations are based on
(50)$$Q=f\left(0,P\right)=\{f\left(0,w_{1}\left(I\left(x,y\right)\right)\right),\ldots ,f\left(0,w_{n}\left(I\left(x,y\right)\right)\right)\},$$
where f(·) is a mapping function.

#### 3.3.3. Deep Learning Architectures

Below, we use the following architectures: VGG-16/VGG-19 [26], MobileNet [27], ResNet-50 [28], Inception v3 [29], Xception [30], and DenseNet-201 [31]. The selection of these architectures was based on the fact that they have been shown to obtain the Top-1 and Top-5 best accuracy and error rates on the Imagenet task proposed for [45], which is an image classification task.

In Table 1, the number of features extracted by each utilized architecture is explained.

The features for each selected architecture are concatenated with the extracted handcraft features, forming a unique feature vector:(51)$$F=D_{eep}\left[\#Features_{Extracted}\right]\cup H_{and}\left[Asymetry,Area,\ldots ,Entropy,Contrast,Std\ldots ,Max,Min,\ldots \right],$$
where the size of vector *F* is equal to 43 for the handcraft features, plus the number of features extracted from the CNN architecture used.

### 3.4. Algorithm Summary

Let the sum of all proposed and explained procedures be described in the form of an algorithm for extraction features for PSL Images. The proposed CAD system consists of four principal stages: (a) preprocessing, (b) handcraft features, (c) deep learning features, and (d) the fusion stage. In the first stage, artifacts are removed by using a Gaussian filter after a CIELab color transformation is employed. Mean thresholding per channel is employed afterwards. We then extract the ROI by using the boundary box algorithm. In the second stage, shape features are extracted using Equations (12)–(17), Equations (18)–(21) are computed for statistical color features, and texture features are then calculated from Equations (22)–(38). The ROI image is finally processed by the chosen CNN Architecture, whose features are concatenated for the following steps.

Algorithm 1 presents the details of the feature extraction process for PSL images.

### 3.5. Feature Selection

After extracting the deep learning and handcraft features, the number of features is reduced, which is a typical problem in machine learning algorithms, because high-dimensional data increases the computing time for a prediction.

Feature selection is one method to resolve this problem. In [46], filtering methods are applied, and the features are selected based on various statistical tests, such as $\chi^{2}$, ANOVA, Linear discriminant analysis (LDA) among others [47].

The extracted data can be represented in the form of a high-dimensional matrix defined by
(52)$$X\in R^{n\times p}$$
where X is the extracted data, *n* represents the instances or elements, and *p* represents the features extracted for each element.

The idea is to reduce the data as much as possible, such that the features are extracted by selecting a subset of each element that is relevant to the pertinence category or label *y*. This subset is defined as
(53)$$X_{S}\in R^{n\times k}$$
where $X_{S}$ represents the reduced data, *n* is the same instance or element of the original data matrix, and *k* is the selected or reduced features based on k<<p.

#### Mutual Information

In this work, we propose the MI metric to reduce the data of the extracted features. MI is a measure based on the entropy measure:(54)$$I\left(X;Y\right)=H\left(X\right)-H\left(X|Y\right)=\sum_{y\in Y}\sum_{x\in X}P\left(X\cap Y\right)log_{2}\left(\frac{P(X\cap Y)}{P(X)P(Y)}\right)$$
where $X=x_{1},\ldots ,x_{n}$ and $Y=y_{1},\ldots ,y_{n}$ in the multi-variable case, H(X|Y) is the conditional entropy between two random variables, and H(X) is the entropy of a random variable [48,49,50].
(55)$$H\left(x\right)=-\sum_{i=1}^{n}P\left(x_{i}\right)log_{2}P\left(x_{i}\right)$$

**Algorithm 1** Algorithm summary
**Require:** PSL Image

   (a)Preprocessing
1:**Input**: *I* Apply Gaussian Filter, Equation (Equation 2)Apply RGB to CIEL*a*b* Color Transform**Separate** CIEL*a*b* Image $I_{Lab}$ in Image Channels $I_{L},I_{a},I_{b}$**Calculate** Mean value of $I_{L},I_{a},I_{b}$, Equations (3)–(5)2:
**for all**
$\left(i,y\right)\in I_{L}$
**do**
3:    **if**
$I_{L}\left(x,y\right)\geq \bar{I_{L}}$
**then**4:        Assign 1 to $I_{Th_{L}}\left(x,y\right)$5:    **else**6:        Assign 0 to $I_{Th_{L}}\left(x,y\right)$7:    **end if**8:
**end for**
9:
**for all**
$\left(i,j\right)\in I_{a}$
**do**
10:    **if**
$I_{a}\left(x,y\right)\geq \bar{I_{a}}$
**then**11:        Assign 1 to $I_{Th_{a}}\left(x,y\right)$12:    **else**13:        Assign 0 to $I_{Th_{a}}\left(x,y\right)$14:    **end if**15:
**end for**
16:
**for all**
$\left(i,j\right)\in I_{b}$
**do**
17:    **if**
$I_{b}\left(x,y\right)\geq \bar{I_{b}}$
**then**18:        Assign 1 to $I_{Th_{b}}\left(x,y\right)$19:    **else**20:        Assign 0 to $I_{Th_{b}}\left(x,y\right)$21:    **end if**22:
**end for**
 **Compute**$I_{bin}\left(x,y\right)$ Applying Equation (Equation 9) to $I_{Th_{L}},I_{Th_{a}},I_{Th_{b}}$    Apply Median Filter, size =5×5**Compute** Bounding Box Algorithm to estimate coordinates of Region Of Interest    Crop $I_{roi}\left(x,y\right)$ from estimated coordinates on I(x,y) & $I_{Lab}\left(x,y\right)$23:**Output**: Region of Interest (ROI) Image $I_{roi}\left(x,y\right)$

    (b)Handcraft Features
24:**Input**: $I_{ROI}$     **Compute** Area, Perimeter, Circularity, Diameter and Eccentricity from Equations (13)–(17)    **Compute** Asymetry from Equations (10)–(12)    **Compute** Color Features from Equations (18)–(21) from $I_{roi}\left(x,y\right)$    **Compute** Texture Features from Equations (22)–(43) from $I_{roi}\left(x,y\right)$25:Concatenate the extracted features *H*26:**Output**: *H* Handcraft features

    (c)Deep Learning features
27:**Input**: $I_{roi}$28:     Load the weights $W_{i}$ from selected CNN architecture29:     Apply the weights $W_{i}$ to $I_{roi}$30:     Obtain the *D* deep learning features31:**Output**: *D* deep learning features 
(D)Wrapping Features
32:**Input**: D,H33:Apply H∪D to the extracted features34:**Output**: *F* Full set of extracted features


Ross, in [51], proposed an MI estimator for continuous and discrete data aiming at the relationship between datasets.

Based on the nearest neighborhood rule, the idea is to find a k-nearest neighbor between a point *i* among all the data points $N_{xi}$ using the Chebyshev distance metric:(56)$$D_{chebyshev}=\max_{i}\left(\right|x_{i}-y_{i}\left|\right),$$
and the MI measure is then computed as follows:(57)$$I\left(X,Y\right)=\langle I_{i}\rangle =\psi \left(N\right)-\langle \psi \left(N_{x}\right)\rangle +\psi \left(k\right)-\langle \psi \left(m\right)\rangle ,$$
where ψ(·) is the digamma function, $\langle N_{x}\rangle$ is the average of the data points, *k* is the number of *k*-closest neighbors to the point *i*, and 〈m〉 is the average counted neighbors among the full dataset.

If our dataset contains continuous data points, these are discretized using a binning method—grouping the data into bins—leading to a binned approximation of MI as follows:(58)$$I\left(X,Y\right)={\langle \log\frac{p(x_{i},b_{i})}{p\left(x_{i}\right)p\left(b_{i}\right)}\rangle }_{i}.$$

After applying the MI method, a new vector is delivered, and it shows the value of MI per concrete feature among all obtained features. Next, the mean value is calculated for the MI among all features. We proposed to use this value as a threshold to discard the features with the lowest MI values and to keep the features with the highest MI values. Table 2 and Table 3 expose examples of different parts of the fused feature vector.
(59)$$L_{features}\left(x,y\right)=F\left(x,y\right)>\bar{M.I(x,y)}$$

The found subset contains the features with the highest mutual information values that we consider as the fused data of both sets of extracted features from a PSL. Below, as an example, we present the behavior of the features and their MI values.

Table 2 and Table 3 expose, for illustration, several MI values between the extracted features for the dataset of images. Some of the deep learning features appear to demonstrate negligible MI values for the binary classification problem, according to the complete set of features extracted from the database, as one can see in Table 2.

Table 3 exposes several features with significant values of MI that are merged with additional significant handcraft features, forming the final set of features for the proposed system. Therefore, the proposed fusion method based on MI measurements demonstrates the fusion for both types of features in accordance with their influence on the binary classification problem.

Therefore, in contrast with state-of-the-art techniques that use Concat, PCA-based, and $\chi^{2}$-based methods, among others, for the selection of significant features, the proposed approach employs the information measures, justifying the informative weight of each feature that is used in the classification stage.

## 4. Results and Discussion

The classifiers: logistic regression, support vector machine with linear and rbf kernel [52], and finally, relevant vector machine [53,54] algorithms were employed in this work. The rationale for the use of different classifiers lies in the fundamental idea of transfer learning, whereby after extracting generic features, a shallow classifier must be applied to test the proposed method.

### 4.1. Experimental Results

#### 4.1.1. Experimental Setup

The described method was performed on a PC with an Intel^®^ Xeon E5 1230-V5 CPU, 32GB RAM, NVIDIA GeForce^®^ 1080Ti with 11 GB RAM, running a Linux 64-bit operating system, Python 3.5, and the libraries: Keras 2.3 [55], Sklearn [56], Mahotas [57], thundersvm [58], and Imblearn [59].

#### 4.1.2. Evaluation Metrics

In this study, we used commonly applied performance metrics: accuracy, sensitivity, specificity, precision, f-score, and the Matthews correlation coefficient:(60)$$Accuracy=\frac{tp+tn}{tp+tn+fp+fn}.$$
The accuracy value measures the appropriate classifications over the total elements evaluated.
(61)$$Sensibility=\frac{tp}{tp+fn}.$$
The sensibility value, also known as recall, measures the number of positive elements that are correctly classified.
(62)$$Specificity=\frac{tn}{tn+fp}.$$
The specificity value measures the number of negative elements that are correctly classified.
(63)$$Precision=\frac{tp}{tp+fp}.$$
The precision value measures the number of elements that are correctly classified among all the positive elements to evaluate.
(64)$$F_{Score}=\frac{2tp}{2tp+fp+fn}.$$
The F-score value measures the harmonic mean between precision and recall.

These criteria are described in terms of tp, tn, fp, and fn, which denote true positive, true negative, false positive, and false negative, respectively. Additionally, to characterize the classifier performance, we used the Matthews correlation coefficient [60]:(65)$$MCC=\frac{\left(tp\times tn)-(fp\times fn\right)}{\sqrt[{2}]{\left(tp+fp)(tp+fn)(tn+fp)(tn+fn\right)}},$$
where the MCC value measures the performance of the classification model as a coefficient between the predicted and the observed elements of the binary classification. It returns a value between [−1,1], where a value of 1 represents a perfect prediction, a value of 0 is no better than a random prediction, and −1 indicates total disagreement between prediction and observation.

#### 4.1.3. Dataset

This study uses the public ISIC 2018: Skin Lesion Analysis Towards Melanoma Detection grand challenge datasets [21] task 3, also known as HAM10000 [61], which contains 10,015 separate images, as shown in Figure 6, where AKIEC corresponds to Actinic Keratosis, BCC is Basal Cell Carcinoma, DF is a Dermatofibroma, MEL is Melanoma, NV is Nevus, BKL is Pigmented Benign Keratosis, and VASC is Vascular. This distribution was obtained from the Ground Truth file, and each image is on RGB space with a size of 450×600.

As one can see in Figure 6, the ISIC dataset contains images that belong to different types of skin lesions that do not correspond to the melanoma-type lesion. Therefore, we decided to modify the dataset to develop a binary class classification by excluding all the classes, except for Melanoma and Nevus.

We split the ISIC dataset: 75% for the training set and 25% for test set. The features extracted were processed by Z-score normalization:(66)$$Z=\frac{x-\mu }{\sigma }.$$

#### 4.1.4. Balance Data

The adjustment of the ISIC dataset mentioned above means that the data are unbalanced, where one class contains more data than others. In most machine learning techniques, when they employ unbalanced data, this can result in a lower performance of the minority class, which can cause misclassification of the data.

SMOTE [62] is a data augmentation method that can oversample the data of the minority class, compensating with the majority class. This method is based on K-NN clustering and Euclidean distance, selecting two points of the minority class and computing a new one based on them. This method is iterative until it reaches an equivalent amount of information of the majority class.

The SMOTE technique has been employed in several studies [63,64], where extracted features that belong to an unbalanced dataset are oversampled to compensate for the number of instances between classes. In this work, we apply this method to compensate the data of the melanoma class against nevus ones to the selected features by the MI criterion, and as result, a balanced dataset with the fused features can be obtained.

The study [65] introduces new metrics to overcome this problem, where geometric mean attempts to maximize the accuracy of each of the two classes similarly, a performance metric that correlates both objectives:(67)$$G_{mean}=\sqrt[{2}]{Sensitivity\cdot Specificity}.$$

Dominance is aimed at quantifying the prevalence relation between the majority and minority classes and is used to analyze the behavior of a binary classifier:(68)Dominance=Sensitivity−Specificity.

The Index of Balanced Accuracy (IBA) is a performance metric in classification that aims to make it more sensitive for imbalanced domains. This metric is defined as follows:(69)$$IBA=\left(1-Dominance\right)\cdot G_{mean}^{2}.$$
The objective of this procedure is to moderately favor the classification models with a higher prediction rate of the minority class, without underestimating the relevance of the majority class.

#### 4.1.5. Experimental Results

In the following tables, the experimental results of binary classification for balance data are presented.

The experimental results in Table 4 show that the designed system appears to demonstrate sufficiently good performance when different CNN architectures are fused with handcraft features in accordance with the MI metric that can seek relevant information among features against concatenating or discriminative analysis.

Table 4 provides the experimental results obtained using the selected CNN architectures, where the Mobilenet v2 appears to demonstrate the best performance in comparison to the before-mentioned architectures. The proposed method shows notable evaluation metrics such as accuracy, Area Under Curve (AUC), and IBA metrics. The selected features contain the fused features that are the most relevant for the classification of a lesion with the MI metric.

Presented in Table 5, experimental results for different criteria show that the designed system outperforms several state-of-the-art methods. The experimentally justified performance, in our view, is due to the fusion technique employed, where a mutual information metric seeks relevant information among features upon concatenating or discriminative analysis. Moreover, the IBA metric is employed, achieving a value of 0.80, which confirms the stability and robustness of the system when balanced data are used.

The proposed method achieves an accuracy of 92.40%, sensitivity of 86.41%, AUC of 89.64%, and an IBA of 0.80. In [24], the authors proposed the usage of the complete PSL image, which contains healthy skin and artefacts. Whereby, there is a probability of generating misclassification of the recognizing patterns that belong to these objects. By contrast, the proposed system extracts the proposed features from the region of interest of an image only for the entire classification process. This guarantees that the feature extraction is performed exactly in the lesion image.

Moreover, our novel CAD employed the ISIC 2018 database [21] that contains more than 10,000 dermoscopy images that are authenticated by experts. Additionally, because of the unbalanced data that are present in this database, we applied the data augmentation procedure given in Section 4.1.4. This guarantees the robustness of the obtained classifications results. In contrast, the CAD DermoDeep system [24] obtained the experimental results using a synthetic database constructed by four different databases (private and public). In this case, an equal number of melanoma and benign skin lesions were subjectively selected from each of the four databases. This system showed slightly improved performance results as those reported in this study. In our opinion, such an approach does not guarantee that the same high performance can be repeated using data that are not previously preselected.

The proposed system used a data augmentation technique and presented the performance analysis for all images contained in the database and not only those that have been preselected according to a subjective criterion that has no statistical justification.

Finally, our proposed system was developed with medical-based and deep learning features, whereby the system employed data from both sets of features and merged them, applying the MI criterion. As a result, the system enhances the recognition of melanoma and nevus lesions compared to the use of a fully deep learning approach that is extremely computationally expensive to train, requires substantial amounts of labeled data and does not recognize dermoscopic features established in the ABCD algorithm.

## 5. Conclusions and Future Work

In this study, a novel competitive CAD system was designed to differentiate melanoma from nevus lesions. Different from commonly proposed CAD systems, the novel method employs handcraft features based on the medical algorithm ABCD rule and deep learning features and applies a transfer learning method as a feature extractor. Additionally, in the proposed system, the set features are fused using an MI metric that, in contrast with state-of-the-art systems, can select the most significant features in accordance with their influence on binary classification decisions.

The performance of the proposed system has been evaluated; the system achieved an accuracy of 92.4%, an IBA of 0.80, and an MCC of 0.7953 using a balanced dataset. The system is competitive against the performance of other, state-of-the-art systems.

The proposed CAD system can help inexperienced physicians to visually distinguish the medical features to be applied. Furthermore, it could be used to provide a second opinion to a dermatologist. Our future work will consist of designing a method for multiclass classification using both sets of features, thus permitting the diagnosis of several diseases found in the ISIC challenge dataset.

## Figures and Tables

**Figure 1 entropy-22-00484-f001:**
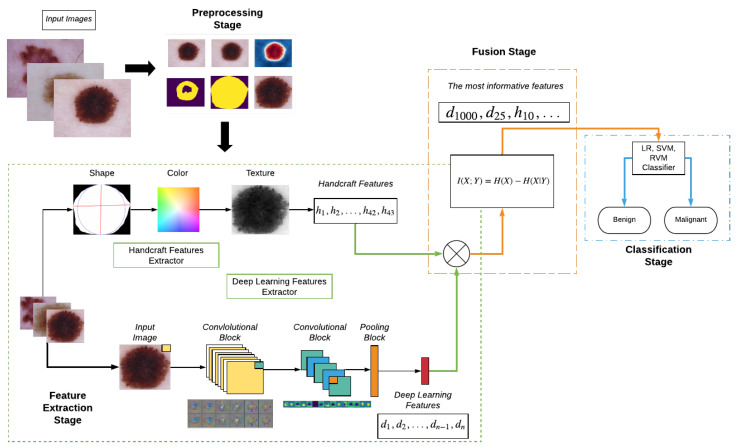
Block diagram of the novel Computer-Aided Detection (CAD) system.

**Figure 2 entropy-22-00484-f002:**
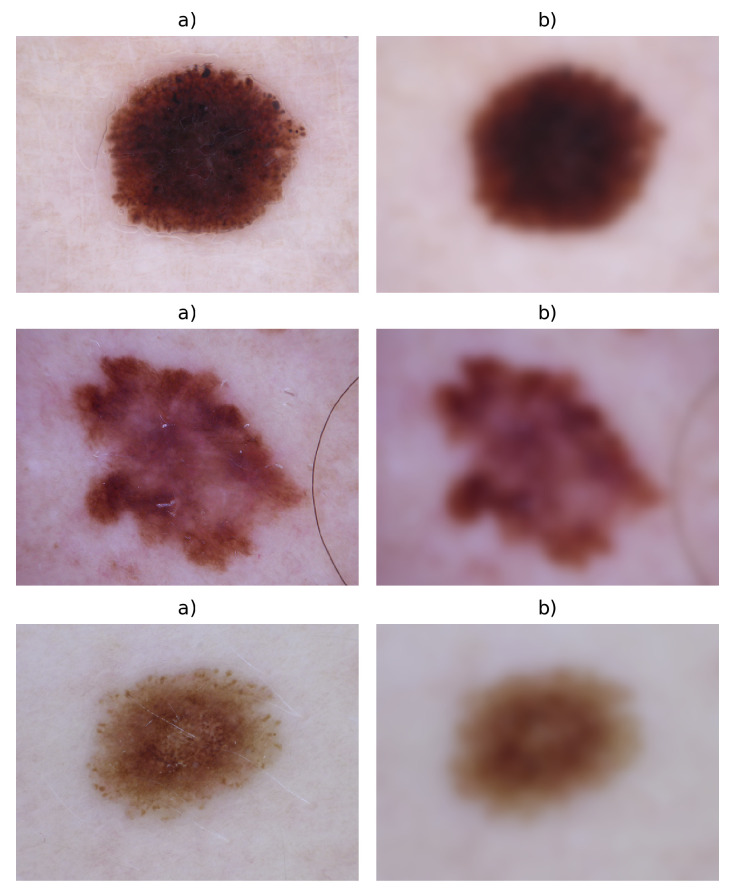
(**a**) Original image I(x,y); (**b**) image (**a**) processed with a Gaussian filter.

**Figure 3 entropy-22-00484-f003:**
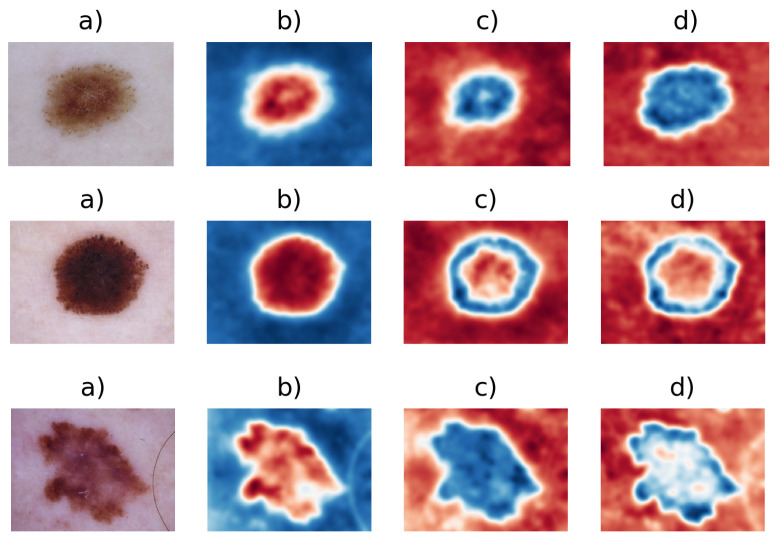
(**a**) Original image I(x,y), (**b**) I(x,y) on channel L, (**c**) I(x,y) on channel a*, and (**d**) I(x,y) on channel b*.

**Figure 4 entropy-22-00484-f004:**
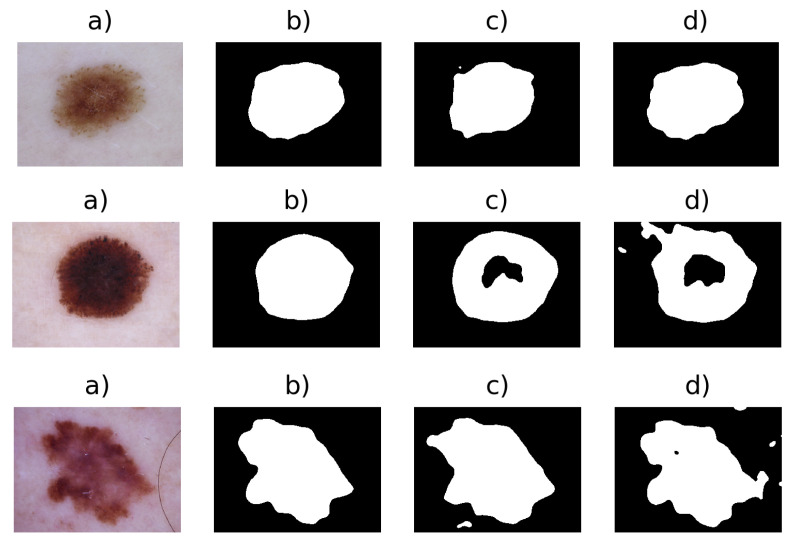
Results of the thresholding stage: (**a**) Original image I(x,y), (**b**) binary image $I_{Th_{L}}$ obtained from the threshold of the L channel, (**c**) binary image $I_{Th_{a}}$ obtained from the threshold of the a* channel, (**d**) binary image $I_{Th_{b}}$ obtained from the threshold of the b* channel.

**Figure 5 entropy-22-00484-f005:**
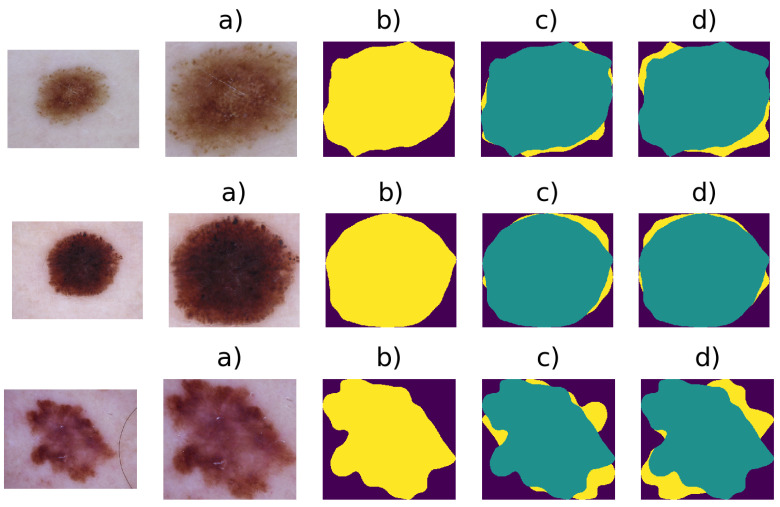
Results of the preprocessing stage: Original image I(x,y), (**a**) Region of Interest (ROI) obtained, (**b**) segmented image $S_{I_{bin}}$, (**c**) obtained asymmetry at 0°, and (**d**) obtained asymmetry at 90°.

**Figure 6 entropy-22-00484-f006:**
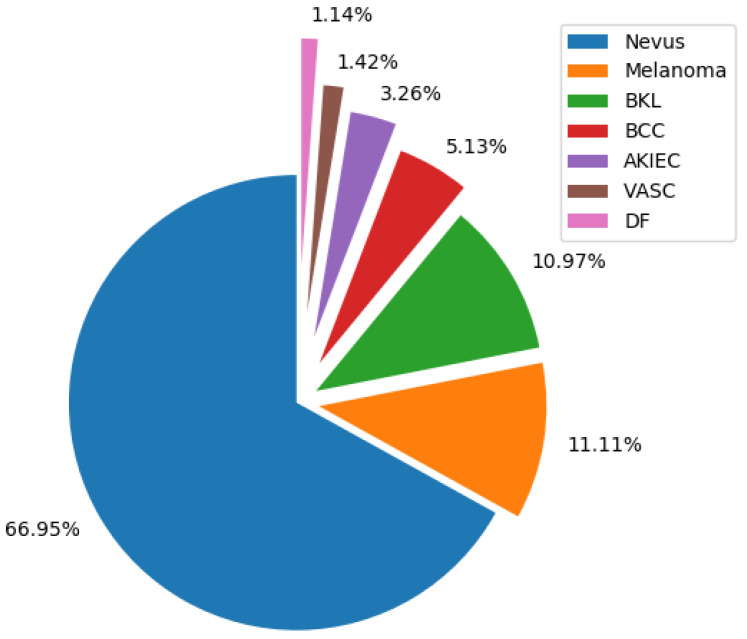
Distribution of classes on the ISIC2018/HAM10000 dataset.

**Table 1 entropy-22-00484-t001:** The number of features extracted by architecture.

CNN Architecture	${\#}_{\mathrm{Features}}$
VGG19	4096
VGG16	4096
ResNET-50	2048
Inception v3	2048
Mobilenet v1	1024
Mobilenet v2	1280
DenseNET-201	1920
Xception	2048

**Table 2 entropy-22-00484-t002:** Example of features that expose the lowest Mutual Information (MI) values.

Inception v3 + Handcraft Features	
*Feature*	*Mutual Info. value*
1448	3.813956037657107 × 10${}^{-7}$
648	1.4314941993776031 × 10${}^{-5}$
1020	3.4236070477255964 × 10${}^{-5}$
804	3.515106075036023 × 10${}^{-5}$
333	4.213506368255793 × 10${}^{-5}$
562	4.4971751217204314 × 10${}^{-5}$
91	5.133156269931938 × 10${}^{-5}$
852	6.514623080855486 × 10${}^{-5}$
1689	7.53828133426282 × 10${}^{-5}$
1788	7.605629690621285 × 10${}^{-5}$

**Table 3 entropy-22-00484-t003:** Example of features that expose the highest MI values.

Inception v3 + Handcraft Features	
*Feature*	*Mutual Info. value*
Mean_b	0.09829037284938669
Min_b	0.09536305749234275
Max_b	0.06834317593527395
578	0.06131147578510121
Min_G	0.05817685924318594
116	0.055293703553799256
389	0.05446628875169646
464	0.05424503079140042
Var_L	0.05420063226575533
288	0.053949117014718606

**Table 4 entropy-22-00484-t004:** Performance results of the proposed method using selected deep learning architectures fused with handcraft features.

	Acc. Train	Acc. Test	Sensibility	Specificity	Precision	F-Score	AUC	G-Mean	IBA	MCC
VGG16	88.60	84.90	79.23	0.85	88.74	83.71	84.79	0.85	0.72	0.7012
VGG19	90.23	87.14	82.46	0.87	90.44	86.26	87.05	0.87	0.76	0.7451
Mobilenet v1	91.48	89.32	84.04	0.89	93.49	88.51	89.21	0.89	0.79	0.7898
Mobilenet v2	**92.40**	**89.71**	**86.41**	**0.90**	**92.08**	**89.16**	**89.64**	**0.90**	**0.80**	**0.7953**
ResNET-50	90.67	87.86	81.24	0.88	93.09	86.76	87.72	0.87	0.77	0.7624
DenseNET-201	91.10	88.54	83.25	0.88	92.61	87.68	88.44	0.88	0.78	0.5985
Inception V3	91.33	88.10	84.87	0.88	90.59	87.42	88.02	0.88	0.77	0.7632
Xception	90.47	87.53	83.19	0.87	90.58	86.73	87.44	0.87	0.76	0.7525

**Table 5 entropy-22-00484-t005:** Comparison between our novel CAD and state-of-the-art CADs.

Metric	[18]	[24]	[22]	Proposed Method Using Mobilenet v2 Architecture
Accuracy	86.07	95	85.55	92.40
Sensibility	78.93	93	86	86.41
Specificity	93.25	-	-	90
Precision	-	93	85	92.08
F-Score	-	-	86	89.16
G-Mean	-	-	-	0.90
IBA	-	-	-	0.80
MCC	-	-	-	0.7953
Imbalance Data	Yes	No	No	No
Fused Data	Yes	Yes	Yes	Yes
Type of Classification	Binary	Binary	Multiclass	Binary

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
