# Peer review of "Melanoma and Nevus Skin Lesion Classification Using Handcraft and Deep Learning Feature Fusion via Mutual Information Measures"

_entropy, 2020, doi:10.3390/e22040484_

Round 1

Reviewer 1 Report

I've started reading this paper with interest, because, despite of some rough English, it seemed to me to be a promising paper, addressing an important problem and proposing an interesting approach. Then I found myself struggling with the text, trying to understand what the authors want to say. And then came tons of equations, some not well explained, motivated or even wrong, although most of them trivial. See, for example, Eq. (2) and the associated explanation that makes no sense...

The most important part, i.e., the one presenting results, discussing them and showing why they make sense, is quite short. In particular, it is very important to demonstrate the robustness of the method, running it with a large number of training/test sampling and with different datasets. Moreover, it seems that the performance of the method is not better than other available methods and, although the authors try to explain why their approach has advantages, the bad writing precludes understanding their reasons. In conclusion, after a promising beginning, I ended up very disappointed.

Reviewer 2 Report

The authors have done considerable work related to the classification of skin lesions. However, the paper is difficult to read for the reasons that would be explained in this review, hoping that it would help.

The very title of the paper is misguiding, as it states “Melanoma classification”, while the abstract broaden the field stating “skin lesion classification”. Then, in line 344, the authors state “We should modify the dataset to develop a binary class classification by excluding all the classes except Melanoma and Nevus.” So it is unclear whether the authors classify Melanoma only, or all skin lesions, or make a binary decision between the melanoma and nevus.

The Abstracts define each abbreviation, e.g. CNN – convolutional neural networks, but it starts with ABCD medical algorithm which is not known to most of the readers. Also, they apply CIELab transformation twice and defined it when the transformation appeared for the third time.

Equation (2) states Iout = 255 x (Iin/255)^gamma, and then they state that gamma = 1. But then Iout=Iin, so how the luminance is normalized?

The method explanations are unsatisfactory. The authors present in detail the mean value calculation (Eq. 4). Gaussian filter (3), z-score, sensitivity and similar features (Eqs 54-59). But, when it comes to Eq (51) for MI that is in the paper title, they state laconically “k is a k-value proposed  in [54]”. However, most readers do know what mean value, sensitivity or z-score are, but “a k-value proposed in [54]” cannot be comprehended at first glance. Besides, considering the same equation (51), it contains notations < y(Nx)> and < y(m)> (averaged digamma functions of Nx and m), but in the explanation below the equation it is said: “< Nx> is the average of the data points, < m>  is the average counted neighbours”. So what is averaged?

The equation that follows, (52), contains the errors. Mutual information is statistically averaged, i.e. the quantity in “averaging brackets” should be multiplied with the joint probability. The denominator contains p(bi), but it should be p(yi).

Although I tried hard, I could not conclude what the authors wish to say with the following sentences and with the equation (53):

By applying this method, it can be obtained a vector headed with the most independent features

to the most dependent of our dataset. So, to perform the subset of features, we apply a threshold based

3on the mean of the vector:

Lf eatures(x, y) = F(x, y) > M.I(x, y) (53)

As the new vector of features is a subset of the features, we

consider these as fusing data between the two types of features extracted.”

Tables 2 and 3 has the same title, but it should be pointed out (in the title) that these are examples of features with high and low MI.

The thresholds (5), (6), (7) are not described at all, it is not clear how did the authors select them. Besides, the meaning of these equations is unclear (the same applies to eq (53)) – or, rather, it is not written mathematically correct. It is not defined what are Ithl, Itha, Ithb.

Bounding box algorithm is given without the reverence or explanation when it first appeared.

It is not clear what the arguments x, y are – spatial coordinates? If so, why they sometimes have indices and sometimes not?

ROI is not defined.

Where is Ich that is used in many equations defined?

The features the authors implement are poorly defined. Sometimes the arguments are x and y, sometimes indexed x and y, sometimes i and j. The feature description should be more thorough, it is intended to inform the reader, not to annoy him!

The entropy formula is written twice: once incorrectly, without the “minus” sign (26), the second time correctly (49).

The language must be improved. “Assuming that both domains are equally”. “the number of features it should be reduced”. “the system should to classify”. “obtenied”.

This is just a sketch of objections considering this paper. The authors need to devote time and effort, to read their work critically, to re-write it and to bring it closer to the readers.  

Reviewer 3 Report

The paper is well written and the method is well described. However, there are 2 concerns:

  1. Authors provide description based on relatively complicated algorithm. It may be difficult to repeat the settings in practice. The difficulty is in many steps of the proposal.
  2. The handcraft or provided by expert data merged with DNN can be valuable impact which is a sound effect of the manuscript. It would be good to refer to the paper "A study in facial features saliency in face recognition: an analytic hierarchy process approach" where a postulation of such approach was given.

I recommend the publication of the paper after correction of the point 2 above.

Round 2

Reviewer 1 Report

The authors made an effort to improve the readability of the paper. Unfortunately, some parts are still badly written. For example, the section reporting the experimental results is confusing. The authors try to explain why is their method better than others, but it is very hard to understand the explanation. Moreover, one of the main concerns issued in the previous review report, namely the lack of a convincing set of experiments (and datasets) showing that a system like this one can be trusted by the physicians, was not properly addressed. For example, the authors say that they use data augmentation methods in order to balance the classes. This is a dangerous operation, when its effect is not completely understood. It is quite different to use data augmentation, for example, in a digit recognition system, where it is "obvious" if, after the transformation, the image still represents the same digit, than when you do not have that guaranty, as in the case of melanoma images. In summary, there are important aspects that need to be solved, besides doing some rewriting.

Reviewer 2 Report

The authors have made a considerable effort to improve the presentation quality of their work. While the original version did show valuable results but inadequately presented, the last version gives quite another impression. The track changes show the work that the authors have performed to improve their contribution, while the track changes turned off reveals a true journal paper.

Reviewer 3 Report

After the corrections the paper should be published. However, I still claim that it is difficult to repeat the whole procedure to compare the results independently.

Round 3

Reviewer 1 Report

I thank the authors for their effort in improving the paper---it is now much better than the first submitted version. I still have points in which I disagree, but I also recognize that the paper is now in a form that may be useful to the community. Hence, I recommend acceptance.